# Synthesis of ZnO Nanoparticles Doped with Cobalt Using Bimetallic ZIFs as Sacrificial Agents

**DOI:** 10.3390/nano10071275

**Published:** 2020-06-30

**Authors:** Vera V. Butova, Vladimir A. Polyakov, Elena A. Erofeeva, Sofia A. Efimova, Mikhail A. Soldatov, Alexander L. Trigub, Yury V. Rusalev, Alexander V. Soldatov

**Affiliations:** 1The Smart Materials Research Institute, Southern Federal University, Sladkova str. 178/24, 344090 Rostov-on-Don, Russia; vlpolyakov@sfedu.ru (V.A.P.); bulanova@sfedu.ru (E.A.E.); sefimova@sfedu.ru (S.A.E.); mikhail.soldatov@gmail.com (M.A.S.); rusalev@sfedu.ru (Y.V.R.); soldatov@sfedu.ru (A.V.S.); 2National Research Centre, Kurchatov Institute, 1 Akademika Kurchatova pl, 123182 Moscow, Russia; alexander.trigub@gmail.com

**Keywords:** ZIF-8, ZIF-67, bimetallic, zinc oxide, cobalt doping, pyrolysis, oxygen vacancies, Co_3_O_4_

## Abstract

We report here a simple two-stage synthesis of zinc–cobalt oxide nanoparticles. We used Zn/Co-zeolite imidazolate framework (ZIF)-8 materials as precursors for annealing and optional impregnation with a silicon source for the formation of a protective layer on the surface of oxide nanoparticles. Using bimetallic ZIFs allowed us to trace the phase transition of the obtained oxide nanoparticles from wurtzite ZnO to spinel Co_3_O_4_ structures. Using (X-ray diffraction) XRD and (X-ray Absorption Near Edge Structure) XANES techniques, we confirmed the incorporation of cobalt ions into the ZnO structure up to 5 mol.% of Co. Simple annealing of Zn/Co-ZIF-8 materials in the air led to the formation of oxide nanoparticles of about 20–30 nm, while additional treatment of ZIFs with silicon source resulted in nanoparticles of about 5–10 nm covered with protective silica layer. We revealed the incorporation of oxygen vacancies in the obtained ZnO nanoparticles using FTIR analysis. All obtained samples were comprehensively characterized, including analysis with a synchrotron radiation source.

## 1. Introduction

Metal–organic frameworks (MOFs) are a class of porous materials with a module structure [1,2]. They are constructed of two parts: inorganic metal clusters and organic molecules [3]. The first ones are called secondary building units, while the second ones are called linkers. The module structure allows the design of frameworks suitable for specific applications and tune functionality to achieve the desired properties [4,5]. This has resulted in the successful use of MOFs in various fields, such as gas storage and separation [6,7,8], catalysis [9], air and water purification [10,11], drug delivery [12], and many others. Recently, MOFs have been widely used for the sacrificial preparation of functional nanoparticles [13,14]. Synthesis of nanostructured materials from MOFs possesses a list of advantages, such as narrow particle size distribution, a simple solid-state decomposition technique, and scalability of the process. Moreover, as MOFs contain organic parts, pyrolysis in inert atmosphere results in porous carbons with well-dispersed nanoparticles. Additionally, the porous structure provides the incorporation of functional species into the cages prior to pyrolysis, which leads to hybrid products.

Zeolite imidazolate frameworks (ZIFs) are a subclass of MOFs with a zeolite topology constructed from metals (Zn, Co, Ni, Cu, and others) and imidazole derivates as linkers [15,16]. ZIFs are extremely attractive for application as sacrificial precursors due to the high concentration of nitrogen in linkers and, therefore, in carbon after pyrolysis. ZIF-8 contains zinc ions coordinated with 2-methylimidazole linkers (Appendix A). ZIF-8 exhibits a high specific surface area (1600 m^2^/g) and thermal stability (up to 400 °C). Zinc ions in ZIF-8 can be partially or entirely substituted with cobalt ions [9]. MOF with ZIF-8 structure constructed from cobalt ions and 2-methylimidazole is usually called ZIF-67. Synthesis of nanostructures from bimetallic ZIFs integrates the high specific surface area and nitrogen content of ZIF-8 with the high degree of graphitization and cobalt content of ZIF-67 [17]. Pyrolysis of a hybrid material containing ZIF-67 and carbon nanotubes leads to the formation of Co–N–C catalyst films suitable for Zn–air batteries as a robust bifunctional air electrode [18]. Bimetallic ZIFs with ZIF-8 structure were used to obtain a hybrid material containing N-doped porous carbon, metallic cobalt, and zinc oxide. This material exhibited high electrocatalytic activity for oxygen reduction [19]. For the same application, a hybrid material obtained by annealing of the ZIF-67 mixture with reduced graphene oxide was tested [20]. Growth of bimetallic Zn/Co-ZIFs on graphene sheets with subsequent pyrolysis allowed the formation of Co/Zn-containing N-doped carbon nanotubes applied for electromagnetic wave absorption [21]. Porous carbon with ZnO/Co_3_O_4_/CoO obtained from bimetallic ZIF-8 exhibited enhanced electrochemical performance as anode for lithium-ion batteries [22]. Additional treatment of ZIFs with silica prevents agglomeration in the pyrolysis process and enhances the biocompatibility of the obtained materials [23].

Annealing of ZIFs in the airflow results in the formation of zinc–cobalt oxides. These materials are of paramount importance due to their use in numerous applications, including catalysis [24], photodynamic therapy [25], antibacterial agents [26], sensing [27,28], and ferromagnetism [29,30].

In the present work, we used bimetallic ZIFs with ZIF-8 structure as a precursor for obtaining zinc oxide nanoparticles doped with cobalt. Initial ZIFs were synthesized using the microwave (MW)-assisted method, which provided nanosized particles of MOFs and enhanced properties of the obtained oxides. Moreover, for part of the samples, we used additional treatment with tetraethoxysilane to decrease the particle size of ZnCo oxides and to form a protective silica cover on the surface of the product. The list of compositions of the initial ZIFs with six variants of Zn/Co ratio allowed us to trace the effect of cobalt content on the structure and properties of the obtained oxides.

## 2. Experimental

### 2.1. Methods

X-ray powder diffraction (XRPD) patterns were measured with a Bruker D2 PHASER X-ray diffractometer (Bruker Corporation, Germany) (CuKα, *λ* = 1.5418 Å) with a step of 0.01°. Jana2006 program package was used for profile analysis [31]. The ratio between wurtzite and spinel phases in synthesized samples was calculated according to the obtained data. We chose two of the most intense reflections: 113 for spinel and 101 for wurtzite. Their exact intensities were estimated during profile analysis, and their ratio indicated that of respective phases. We used two methods to calculate the average size of the particles. The first one was according to the Sherrer equation. We used five reflections for wurtzite-type phases (102, 2–10, 103, 2–12, 201) and five reflections for spinel components (113, 222, 004, 115, 404). A standard quartz sample was used to distinguish the instrumental contribution to the peak width in the calculations according to the Scherrer equation. For Williamson–Hall analysis in Jana2006, all observed reflections were considered (17 for wurtzite and 18 for spinel) to plot a straight line according to the equation β cos θ = kλ/D + 4ε sin θ. The intercept gave the average particle size of the sample.

The M4 TORNADO was used for elemental analysis by applying small-spot micro X-ray fluorescence (Micro-XRF) analysis. IR spectra were measured on a Bruker Vertex 70 spectrometer in ATR (attenuated total reflectance) geometry with an MCT detector and a Bruker Platinum ATR attachment. The spectra were measured in the range of 5000 to 300 cm^−1^ with a resolution of 1 cm^−1^ and 128 scans. The reference was air. High-resolution transmission spectra were measured with the FEI Tecnai G2 F20 (Hillsboro, Oregon, United States) microscope with EDAX Apollo XLT EDS detector (AMETEK Co., Tokyo, Japan) (accelerating voltage 200 kV). Nitrogen adsorption isotherms were measured on ASAP2020 (Micromeritics, Georgia, USA) equipment at −196 °C. All samples prior to measurement were degassed at 150 °C for 8 h in a dynamic vacuum. Magnetic measurements were carried out on a Lakeshore VSM 7404 magnetometer (LakeShore Cryotronics, Inc., Westerville, OH, USA). Magnetization curves were measured at room temperature in the field range of −19 to 19 kOe. Each measurement had at least 160 points with a shutter speed of 10 s per point.

Spectra of 99Zn1Co-T and 0Zn100Co-T as well as reference Co_3_O_4_ and CoO spectra were measured at the structural material science beamline of Kurchatov synchrotron. The X-ray beam was monochromatized by means of Si(111) monochromator in transmission geometry using N_2_-filled ionization chambers. The experimental spectra were normalized using standard procedures by means of Athena software [32]. Theoretical simulations of XANES spectra were performed by means of finite difference method using FDMNES software [33,34,35].

### 2.2. Synthesis

Reagents Zn(NO_3_)_2_·6H_2_O, Co(NO_3_)_2_·6H_2_O, triethylamine (TEA), dimethylformamide (DMF), 2-methyl imidazole, and tetraethoxysilane (TEOS) were purchased from Sigma-Aldrich (St. Louis, Missouri, USA). Ultrapure water (18 MΩ·cm) was produced by SimplicityUV (Millipore, Nihon Millipore, Japan) from distilled water.

As precursors for annealing, we used Zn/Co-ZIF-8 samples synthesized according to a previously reported technique [9]. Briefly, zinc and cobalt nitrate hexahydrate, 2-methyl imidazole, and triethylamine were dissolved in DMF with molar ratio 1:4:2.6:289. We mixed Zn and Co in the proportions 1:0, 0.99:0.01, 0.95:0.05, 0.75:0.25, 0.5:0.5, and 0:1 (Appendix A). The obtained solution was closed hermetically and heated in the MW oven at 140 °C for 15 min with magnetic stirring. After cooling down to room temperature, a precipitate was collected by centrifugation, washed, and dried.

Each Zn/Co-ZIF-8 sample was split into two parts. The first one was used as is. The other one was additionally treated as follows: 300 mg of Zn/Co-ZIF-8 sample was mixed with 5 mL of TEOS at room temperature for two hours (Figure 1). Then, the powder was collected using centrifugation, washed with methanol, and dried in the air. For annealing, 0.05 g of each sample was heated at 500 °C in the air for two hours. The samples are designated according to the Zn/Co molar ratio in the ZIF precursors, while samples obtained with TEOS are pointed by letter T (Table 1).

## 3. Results and Discussion

### 3.1. XRD

All ZIF precursors had ZIF-8 structure (Appendix A). XRD powder patterns of the obtained oxide samples are presented in Figure 2. According to this, the increase in cobalt content in the oxide materials resulted in a phase transition from the wurtzite structure of ZnO to the spinel structure of Co_3_O_4_ (Appendix A).

As Zn/Co oxides are extremely attractive for many applications, a lot of compositions have been reported. One part of these oxides can be formed by partial substitution of Zn^2+^ ions in tetrahedral positions of wurtzite by Co^2+^ ions [29]. This leads to a hexagonal wurtzite structure. The other option is the substitution of Co^2+^ ions in the spinel structure by Zn^2+^ ions [36]. This enables the formation of a cubic spinel framework, where Co^3+^ ions occupy octahedral positions, while Zn^2+^ and Co^2+^ are distributed along with tetrahedral sites.

In our experiment, we observed that up to 5 mol.% of Co^2+^ could be incorporated into the wurtzite structure. Sample 75Zn25Co contained a little admixture of the Co_3_O_4_ spinel. In 50Zn50Co, we observed two phases, while 0Zn100Co formed a pure Co_3_O_4_ phase.

Samples obtained with TEOS additive exhibited broad reflections on XRD profiles. This can be attributed to the small size of the particles. The average size of the particles was calculated according to the Sherrer equation as well as using the Williamson–Hall method (Appendix A). The obtained values are provided in Table 2. As can be observed, the size of the particles obtained with TEOS additive was at least half that of the analog composition obtained without TEOS. The increase in cobalt concentration resulted in phase transition as well. However, the sample 50Zn50Co-T contained only the spinel phase according to XRD, while the same composition without TEOS additive resulted in a mixture of wurtzite and spinel phases.

Unit cell parameters were determined using the Jana2006 program package. We revealed the following trend: TEOS additive during the synthesis provided a small decrease in hexagonal lattice constant *a* and increase in parameter *c*. This indicated elongation of the ZnO_4_ tetrahedra in the wurtzite structure along crystallographic axis *c*.

### 3.2. TEM

ZIF precursors contained nanoparticles of about 50–100 nm with square and hexagonal shape (Appendix A). TEM images of samples obtained without TEOS additive revealed nanoparticles with size 20–30 nm (Figure 3 and Figure 4, Table 2). Hexagonal nanoparticles with well-defined morphology did not exhibit strong aggregation. Each sample contained big particles (about 60 nm) as an admixture to the main fraction of small particles (30 nm) (Figure 4). An increase in cobalt content led to phase transition from wurtzite-like phase (100Zn0Co) to spinel-like phase (0Zn100Co) according to XRD data. Samples 100Zn0Co and 99Zn1Co exhibited hexagonal particles in good agreement with the hexagonal symmetry of wurtzite. Conversely, sample 0Zn100Co with spinel cubic structure was crystallized in the shape of elongated sheets with a distorted hexagonal shape. This can be attributed to crystallization in the (110) crystal plane [37]. However, as the shape of particles of both wurtzite and spinel phases was close to hexagonal, we applied two-dimensional fast Fourier transformation (FFT) to high-resolution TEM images for phase identification. The results are provided in Figure 3 and Appendix A. The distances of adjacent planes in sample 100Zn0Co were about 0.25, 0.26, and 0.28 nm, which corresponded to the distances (110), (200), and (010) planes of the ZnO wurtzite structure. For the 0Zn100Co sample, we observed d-spacing of about 0.46 and 0.24 nm, corresponding to (111) and (311) planes. For the 99Zn1Co sample, d-spacing of 0.26 and 0.28 nm could be indexed as (010) and (200) planes, which confirmed the incorporation of cobalt ions to the wurtzite structure.

Samples obtained with TEOS admixture formed aggregates of small nanoparticles with an average size of about 5–7 nm (Figure 4). TEM images of particles did not exhibit well-shaped crystals. Smoothed edges could indicate silica shell on the surface of oxide nanoparticles. EDX (Energy-dispersive X-ray spectroscopy) mapping confirmed the distribution of Si along with Zn/Co (Appendix A). FFT analysis of high-resolution TEM images showed a pure Co_3_O_4_ spinel phase in the sample 0Zn100Co-T (Appendix A). The distances of adjacent planes in sample 95Zn5Co-T were in a good agreement with planes of the ZnO wurtzite structure (Appendix A). This confirmed the successful incorporation of cobalt ions into the wurtzite-type structure. Sample 100Zn0Co-T, along with expected ZnO wurtzite-type reflections, contained additional phases associated with zinc silicates. As we did not observe these reflections on powder XRD profiles, we suggest that these silicates were formed as a thin layer on the surface of nanoparticles.

### 3.3. XANES

Figure 5a shows XANES spectra measured for 95Zn5Co-T and 0Zn100Co-T samples compared to reference CoO and Co_3_O_4_ compounds. The rising edge position and the shape of the spectrum for the 0Zn100Co-T sample were in a good agreement with those of the Co_3_O_4_ spectrum (a 1:2 mixture of Co^2+^ and Co^3+^, respectively) (Figure 5b). However, there was an evident high energy shoulder of the main peak *B* on the spectrum of sample 0Zn100Co-T. This is a signature of a similar oxidation state and local coordination with a modest variation of cell parameters. Moreover, this can be explained as a mixture of the contributions from the core and shell structure, where the cell parameters of the “core” are similar to pure Co_3_O_4_ and the “shell” of the Co_3_O_4_ structure with smaller cell parameters that come from shortened interatomic distances.

The rising edge in the spectrum for the 95Zn5Co-T sample showed an evident energy shift to low energies compared to 0Zn100Co-T and Co_3_O_4_ reference and even compared to CoO. At the same time, the pre-edge intensity for the 95Zn5Co-T sample had the highest intensity compared to the spectra of reference compounds, suggesting different local coordination for Co. High intensity of the pre-edge feature on the XANES spectra of 3d metals could be a signature of tetrahedral coordination [38]. In order to identify the local coordination of Co ions in the 95Zn5Co-T sample, XANES spectra for a set of model structures were simulated (Figure 6).

The XANES spectra were simulated for a set of model structures. In order to simulate the CoO spectrum, a structure of CoO was taken from the Open Crystallography database (#9008618). For Co_3_O_4_ (Open Crystallography database #1538531), a spectrum for both tetrahedral and octahedral sites was simulated and summed up with 1:2 weights. In order to simulate tetrahedral Co coordination, the two models were built based on the ZnO wurtzite phase (Open Crystallography database #9008877). In the first “CoO cubic ZnO structure” model, all Zn atoms from the structure were substituted with cobalt atoms without any relaxation, and the simulation was performed for the periodic structure. For the second “Co single atom in ZnO” model, the calculations were performed in the direct space for the cluster of ZnO of 8 Å radius with a Co substitution in the center of the cluster.

The simulated spectra for the structures of the reference compounds were in reasonable agreement with the experiment and reproduced the main spectroscopic trends. The spectra simulated for both models of the tetrahedral Co coordination were in good agreement with experimental data, confirming tetrahedral coordination in the 95Zn5Co-T sample.

### 3.4. Magnetic Properties

After subtracting the diamagnetic background of the cell, it was observed that hysteresis loops did not reach saturation (Appendix A). The coercive force for samples 100Zn0Co-T, 50Zn50Co-T, and 0Zn100Co-T was about 20 Oe. However, for samples obtained without TEOS, we observed one outstanding point. The coercive force for sample 0Zn100Co was much greater than for samples 100Zn0Co and 50Zn50Co, namely, 87 Oe for sample 0Zn100Co and ~20 Oe for samples 100Zn0Co and 50Zn50Co. For the samples synthesized without TEOS, an increase in cobalt content resulted in an increase in the paramagnetic contribution. Paramagnetic contribution was possible from very small nanoparticles, which could be superparamagnets. In addition, with an increase in the concentration of cobalt, we found an increase in the total magnetic moment.

### 3.5. FTIR

FTIR spectra of the samples are presented in Figure 7. The spectra of samples 100Zn0Co and 100Zn0Co-T contained a peak at 505 cm^−1^, which can be attributed to zinc oxide with oxygen vacancies [39]. The same peak could be observed on the spectrum of the 50Zn50Co sample, which contained a mixture of wurtzite and spinel structures according to XRD (Table 2). The formation of oxygen vacancies could be associated with the ZIF-8 precursor, which did not contain oxygen. The only source of the last one was air. However, gaseous by-products obtained from the interaction of linkers with oxygen could have reduced its concentration in the reaction area. As a result, synthesized ZnO contained defects in the framework related to oxygen vacancies. In the case of cobalt oxide, the same conditions could have led to a partial reduction of Co_3_O_4_. We did not observe the CoO phase on XRD patterns (Figure 2). Therefore, we propose the formation of a partially reduced Co_3_O_4_ phase as a defect component or on the surface of the particles. A shoulder at 505 cm^−1^ observed on spectra of 0Zn100Co and 0Zn100Co-T samples can be attributed to this phase [40]. Modes at 660, 590, and 555 cm^−1^ could be assigned to vibrations of Co–O bonds in Co_3_O_4_. A peak at 660 cm^−1^ is often associated with vibrations of Co^2+^ ions in tetrahedral coordination, while modes in 550–590 cm^−1^ range rise from Co^3+^ vibrations in octahedral coordination [41,42].

### 3.6. Nitrogen Adsorption

Nitrogen adsorption–desorption isotherms for the samples 100Zn0Co, 100Zn0Co-T, 50Zn50Co, 50Zn50Co-T, 0Zn100Co, and 0Zn100Co-T are presented in Figure 8. All of them exhibited hysteresis loops due to capillary condensation into spaces between nanoparticles. It could be observed that samples obtained with TEOS additive showed higher nitrogen capacities. The specific surface areas for these samples as well as the ZIF-precursors were calculated according to the BET (Brunauer–Emmett–Teller) model (see Appendix A). Samples obtained without TEOS had specific surface areas in the range of 11–18 m^2^/g (see Appendix A). Samples 100Zn0Co-T, 50Zn50Co-T, and 0Zn100Co-T composed from the smaller particles and in good agreement with it exhibited higher BET values in the range 40–100 m^2^/g (see Appendix A). Pore size distribution was calculated for those samples according to the BJH (Barrett-Joyner-Halenda) model using desorption branches of isotherms (Figure 8). In all cases, we observed the same trend. Samples obtained with TEOS additive exhibited smaller pores of about 20 nm, while samples 100Zn0Co, 50Zn50Co, and 0Zn100Co had double the pore size of about 40 nm. These pores were formed as cages between particles in agglomerates. Therefore, it indicated a smaller size of oxide particles of samples 100Zn0Co-T, 50Zn50Co-T, and 0Zn100Co-T in comparison to those obtained without TEOS.

## 4. Conclusions

We have reported the successful synthesis of zinc–cobalt oxide nanoparticles from ZIF precursors. At the first stage, we obtained porous materials with a ZIF-8 structure. They contained Zn and Co ions in various ratios. We applied the MW-assisted synthesis technique, which allowed us to obtain ZIF precursors as nanoparticles with size 50–100 nm and a high specific surface area. The application of such precursors for the synthesis of oxide nanoparticles led to some essential features. First of all, Zn^2+^ and Co^2+^ ions were mixed in the structure of the ZIF precursor, which allowed us to trace all the steps of doping. We observed that pure zinc ZIF-8 after annealing formed the wurtzite structure of ZnO. ZIF precursors with 1 and 5% of Co after annealing produced products with the same structure. Therefore, cobalt ions in the obtained oxides substituted zinc in a hexagonal wurtzite structure. We confirmed this with XRD and XANES analysis. It should be mentioned that a higher concentration of cobalt in the ZIF precursor led to the admixture of the cubic spinel phase of Co_3_O_4_ in the product. ZIF-67 material decomposed to Co_3_O_4_ nanoparticles with a spinel structure. The next feature was related to the crystal structure of ZIF precursors. Simple annealing of these materials in the air resulted in zinc–cobalt oxide nanoparticles of about 20–30 nm. ZnN_4_ clusters in ZIF-8 were separated from each other by organic linkers. This decreased the aggregation process during the formation of zinc–cobalt oxide nanoparticles. Moreover, according to FTIR analysis, oxygen vacancies were incorporated to the obtained oxides with a wurtzite structure, while spinel-type Co_3_O_4_ nanoparticles contained partially reduced cobalt. This could be associated with the initial coordination of Zn/Co ions by nitrogen in the ZIF precursor and low oxygen concentration in the reaction area due to the formation of gaseous products during the interaction of organic linkers with oxygen. Finally, the porous structure of ZIF precursors allowed the introduction of TEOS molecules by their impregnation before annealing. This led to the formation of silicon oxide during heating. Its layer prevented aggregation of oxide nuclei and reduced the size of the particles of Zn/Co oxides down to 5–10 nm. This was revealed with the help of the broadening of diffraction peaks, increased specific surface areas, and TEM images. We conclude that ZIF precursors can be considered for the synthesis of oxide nanoparticles, both with wurtzite and spinel structures.

## Figures and Tables

**Figure 1 nanomaterials-10-01275-f001:**
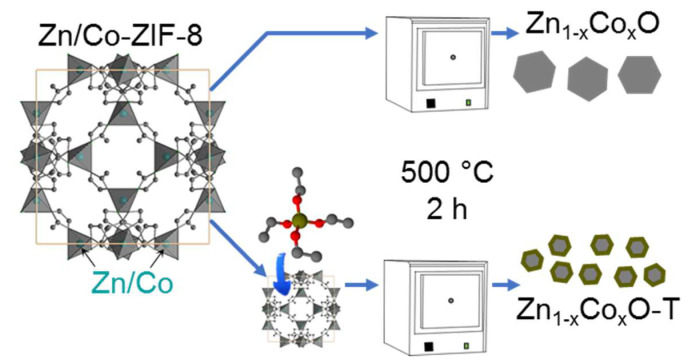
Synthesis of oxide nanoparticles from bimetallic Zn/Co-zeolite imidazolate framework (ZIF)-8 materials.

**Figure 2 nanomaterials-10-01275-f002:**
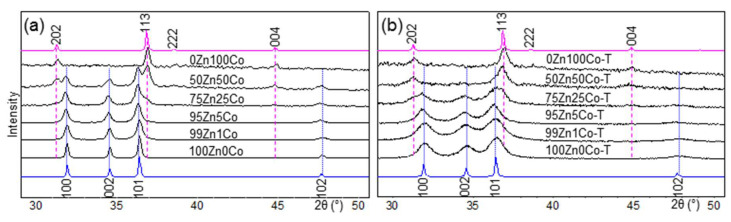
XRD patterns of the obtained oxides without tetraethoxysilane (TEOS) additive (**a**) and with it (**b**). Blue profiles in the bottom represent ZnO wurtzite structure, while pink profiles at the top of the figure correspond to the Co_3_O_4_ spinel structure. ZnO and Co_3_O_4_ profiles were calculated according to crystallographic data from COD sample-ID 2107059 and 1548825, respectively.

**Figure 3 nanomaterials-10-01275-f003:**
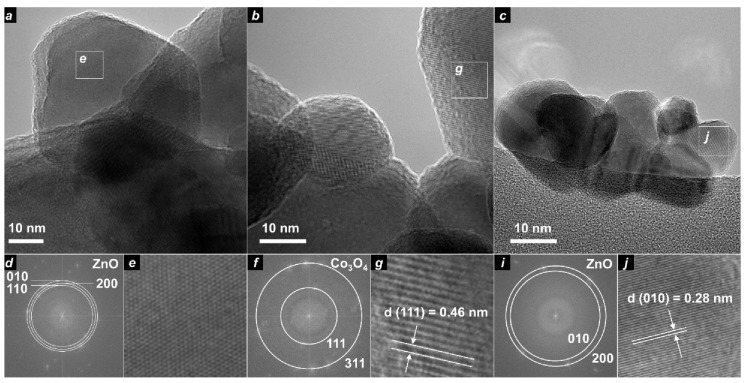
TEM images of samples 100Zn0Co (**a**), 0Zn100Co (**b**), and 99Zn1Co (**c**). On each part, a square region was chosen, and it is provided below with magnification (**e**,**g**,**j**). For these regions, d-spacing was determined (**d**,**f**,**i**) using fast Fourier transformation (FFT) analysis.

**Figure 4 nanomaterials-10-01275-f004:**
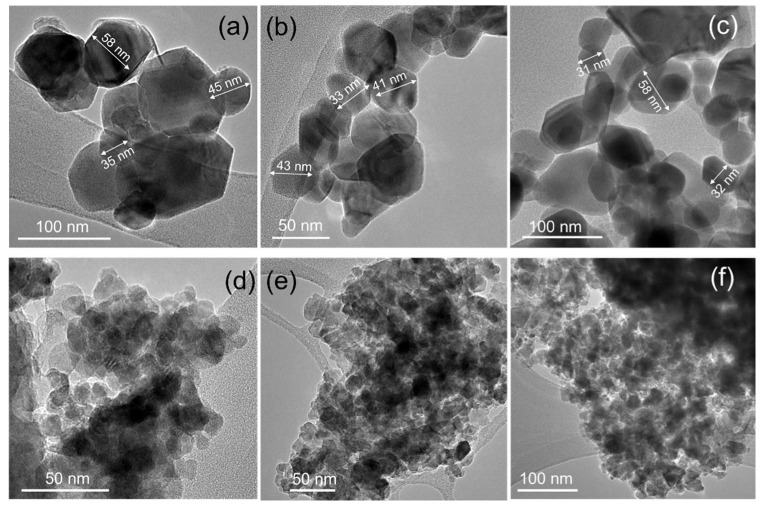
TEM images of synthesized samples 100Zn0Co (**a**), 99Zn1Co (**b**), 0Zn100Co (**c**), 100Zn0Co-T (**d**), 95Zn5Co-T (**e**), and 0Zn100Co-T (**f**).

**Figure 5 nanomaterials-10-01275-f005:**
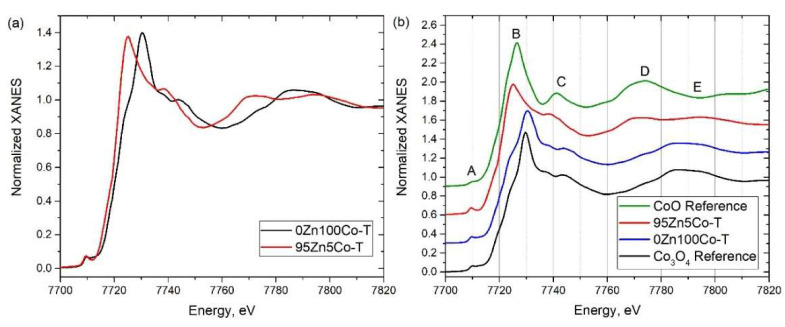
(**a**) XANES spectra measured for 95Zn5Co-T and 0Zn100Co-T samples. (**b**) XANES spectra measured for 95Zn5Co-T and 0Zn100Co-T samples compared to reference CoO and Co_3_O_4_ compounds.

**Figure 6 nanomaterials-10-01275-f006:**
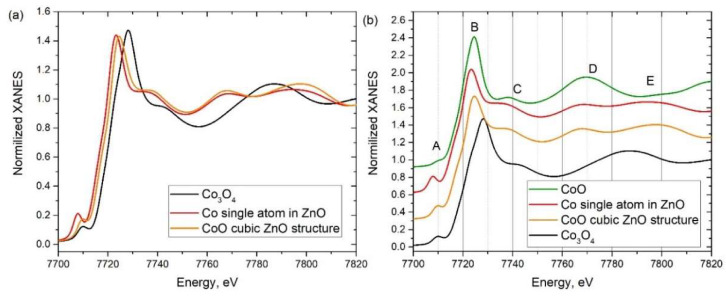
(**a**) XANES spectra simulated for the model structure of Co_3_O_4_ and ZnO with Co substitution. (**b**) XANES spectra simulated for model Co_3_O_4_, CoO, and ZnO with Co substitution.

**Figure 7 nanomaterials-10-01275-f007:**
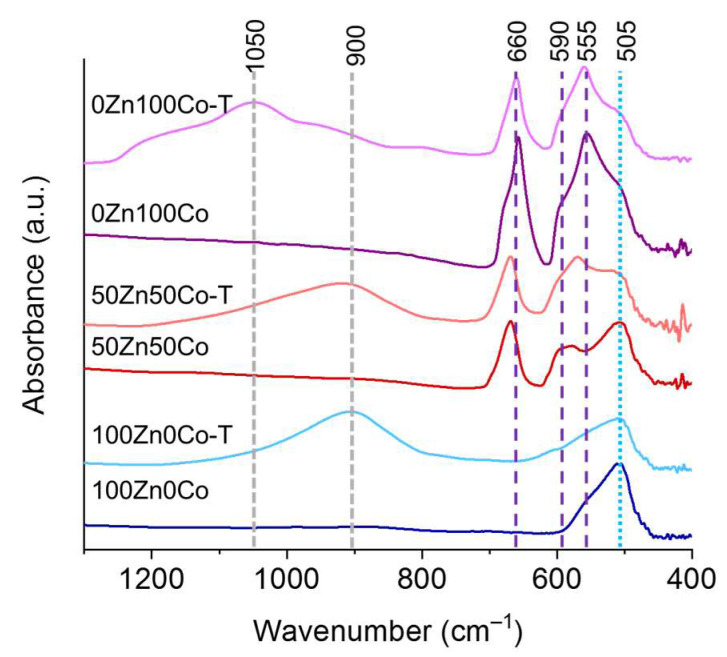
FTIR spectra of samples 100Zn0Co (blue), 100Zn0Co-T (light blue), 50Zn50Co (red), 50Zn50Co-T (light red), 0Zn100Co (purple), and 0Zn100Co-T (light purple).

**Figure 8 nanomaterials-10-01275-f008:**
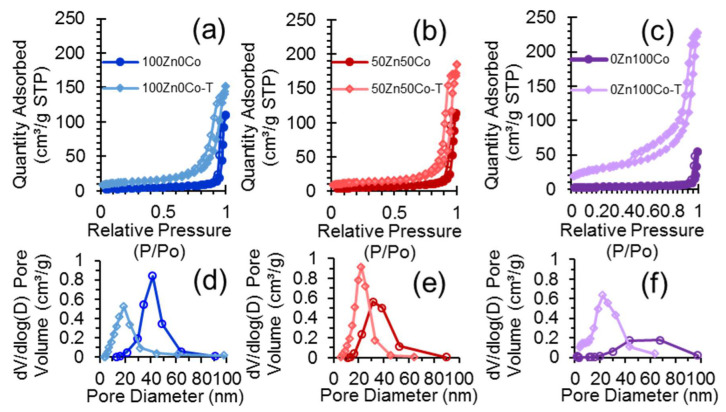
Nitrogen adsorption–desorption isotherms measured for samples 100Zn0Co and 100Zn0Co-T (**a**), 50Zn50Co and 50Zn50Co-T (**b**), and 0Zn100Co and 0Zn100Co-T (**c**). Filled markers correspond to adsorption branches of isotherms, while empty markers indicate desorption ones. The bottom part represents pore size distribution according to the BJH model for samples 100Zn0Co and 100Zn0Co-T (**d**), 50Zn50Co and 50Zn50Co-T (**e**), and 0Zn100Co and 0Zn100Co-T (**f**).

**Table 1 nanomaterials-10-01275-t001:** Sample designation and some synthesis details.

Sample Designation	Molar Ratio Zn:Co	Precursor	Impregnation with TEOS
100Zn0Co	1:0	Zn(C_4_N_2_H_5_)_2_	no
100Zn0Co-T			yes
99Zn1Co	0.99:0.01	Zn_0.99_Co_0.01_(C_4_N_2_H_5_)_2_	no
99Zn1Co-T			yes
95Zn5Co	0.95:0.05	Zn_0.95_Co_0.05_(C_4_N_2_H_5_)_2_	no
95Zn5Co-T			yes
75Zn25Co	0.75:0.25	Zn_0.75_Co_025_(C_4_N_2_H_5_)_2_	no
75Zn25Co-T			yes
50Zn50Co	0.5:0.5	Zn_0.5_Co_0.5_(C_4_N_2_H_5_)_2_	no
50Zn50Co-T			yes
0Zn100Co	0:1	Co(C_4_N_2_H_5_)_2_	no
0Zn100Co-T			yes

**Table 2 nanomaterials-10-01275-t002:** The phase composition and some properties of the obtained Zn/Co oxides. Phase composition is designated as “W” for wurtzite structure and “S” for spinel one. Particle size was calculated according to XRD data using the Sherrer equation (designated as Sh) and Williamson–Hall method (designated as W-H).

Sample Designation	XRF (at.%)	Phase (%)	Unit Cell Parameters (Å)	Particle Size (nm)
Zn	Co	Si		*a*	*c*	Sh	W-H	TEM
100Zn0Co	100	-	-	W	3.24872(8)	5.20630(14)	31.16	29.7	34
100Zn0Co-T	90.5	-	7.4	W	3.2477(3)	5.2084(6)	10.24	6.7	5.6
99Zn1Co	99.3	0.7	-	W	3.24738(10)	5.20518(18)	27.91	23.5	18.2
99Zn1Co-T	93.3	0.6	6.0	W	3.2449(4)	5.2100(7)	7.93	5.3	6.7
95Zn5Co	96.6	3.4	-	W	3.2487(2)	5.2055(4)	22.45	15.3	
95Zn5Co-T	91.7	3.3	4.8	W	3.2472(3)	5.2069(6)	9.19	8.1	5.9
75Zn25Co	78.2	21.8	-	W(29)	3.25016(16)	5.2063(3)	23.15	20.1	-
S(71)	8.0992(5)	16.86	18.2
75Zn25Co-T	74.0	17.9	8.1	W(30)	3.2478(6)	5.2099(10)	7.24	6.3	6.7
S(70)	8.0991(13)	12.7	10.8
50Zn50Co	50.9	49.1	-	W(46)	3.2530(3)	5.2047(5)	29.4	19.8	
S(53)	8.0993(6)	26.65	18.6
50Zn50Co-T	50.1	40.2	9.7	S	8.0998(11)	15.76	12.9	5.5
0Zn100Co	0	100	-	S	8.0968(9)	32.2	25.5	
0Zn100Co-T	0	87.4	12.6	S	8.0811(12)	21.9	20.9

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
