# Peer review of "Synthesis of ZnO Nanoparticles Doped with Cobalt Using Bimetallic ZIFs as Sacrificial Agents"

_nanomaterials, 2020, doi:10.3390/nano10071275_

Round 1

Reviewer 1 Report

The manuscript proposed by Butova et al. deals with the synthesis of ZnO particles doped with cobalt. It is well-written and proposed a synthesis protocole to obtain such nanoparticles and an effective characterization (coupling different techniques such as XRD, TEM,...).

According to the referee, the article is suitable for publication in Nanomaterials after few minor clarifications:

1) some english modifications are required: for instance p2 line 57 "This materials" should be modified into "These materials"

2) The use of Scherrer law to determine the size of the particle should be clarified in the case of broader peaks obtained for TEOS-treated samples. However it is right that the comparison with TEM measurements show a good agreement.

3) There is no XANES results for samples obtained without TEOS. It could interesting to have a comparison for samples with and without use of TEOS. Indeed it could fine to understand the impact of TEOS (if any) on the XANES results.

4) Some additional information should be given by the authors for the determination of the different phases proportion using JANA-2006.

Author Response

Reviewer 1

The manuscript proposed by Butova et al. deals with the synthesis of ZnO particles doped with cobalt. It is well-written and proposed a synthesis protocole to obtain such nanoparticles and an effective characterization (coupling different techniques such as XRD, TEM,...).

The authors appreciate a detailed revision of the manuscript and thank Reviewer 1 for the positive feedback of the work and helpful comments.

According to the referee, the article is suitable for publication in Nanomaterials after few minor clarifications:

1) some english modifications are required: for instance p2 line 57 "This materials" should be modified into "These materials"

Thank you. We fixed this misprint. And we double-checked all text to improve English.  

2) The use of Scherrer law to determine the size of the particle should be clarified in the case of broader peaks obtained for TEOS-treated samples. However it is right that the comparison with TEM measurements show a good agreement.

Details were added to section Experiment:

To calculate particle size, we used two methods. The first one was according to the Sherrer equation. We used five reflections for wurtzite-type phases (102, 2-10, 103, 2-12, 201) and five reflections for spinel-components (113, 222, 004, 115, 404).  A standard quartz sample was used to distinguish the instrumental contribution to the peak width in calculations according to the Scherrer equation. For Williamson-Hall analysis in Jana2006, all observed reflections were considered (17 for wurtzite and 18 for spinel) to plot straight line according to equation βcosθ = kλ/D + 4εsinθ. The intercept gave the average particle size of the sample.

3) There is no XANES results for samples obtained without TEOS. It could interesting to have a comparison for samples with and without use of TEOS. Indeed it could fine to understand the impact of TEOS (if any) on the XANES results.

It is quite interesting to compare the results with and without TEOS from a general point of view. However, the Si-containing samples are of the main interest due to the smaller size of particles. That is why we limit discussion to the samples that were synthesized with TEOS, while samples obtained without TEOS are out of the scope of this study and will be done in the separate paper.

4) Some additional information should be given by the authors for the determination of the different phases proportion using JANA-2006.

The proportions were determined and added to Table 2. Details of calculations are provided in part Methods.

Reviewer 2 Report

The manuscript entitled “Synthesis of ZnO Nanoparticles Doped with Cobalt using Bimetallic ZIFs as Sacrificial Agents” has described the two-stage synthesis of Zn-Co oxide nanoparticles and Zn/Co-ZIF-8 materials as precursors for annealing. The synthesized materials have been characterized using XRD and XNES techniques to confirm the impregnation of cobalt ions into the ZnO structure. This research work is good and will be beneficial for the researchers working in this field. The present manuscript could be accepted after addressing the below comments/suggestions.

They are:

  1. Please check the manuscript for typographical errors.
  2. Some relevant references on metal organic frameworks are missing and should be cited in the main body of the manuscript (Meng, J., Liu, X., Niu, C., Pang, Q., Li, J., Liu, F., Liu, Z. and Mai, L., 2020. Advances in metal–organic framework coatings: versatile synthesis and broad applications. Chemical Society Reviews; Verma, S., Baig, R.N., Nadagouda, M.N. and Varma, R.S., 2016. Titanium-based zeolitic imidazolate framework for chemical fixation of carbon dioxide. Green Chemistry, 18(18), pp.4855-4858; Aguiar, L.W., Otto, G.P., Kupfer, V.L., Fávaro, S.L., Silva, C.T., Moisés, M.P., de Almeida, L., Guilherme, M.R., Radovanovic, E., Girotto, E.M. and Rinaldi, A.W., 2020. Simple, fast, and low-cost synthesis of MIL-100 and MIL-88B in a modified domestic microwave oven. Materials Letters, p.128127.)

Author Response

Reviewer 2

The manuscript entitled “Synthesis of ZnO Nanoparticles Doped with Cobalt using Bimetallic ZIFs as Sacrificial Agents” has described the two-stage synthesis of Zn-Co oxide nanoparticles and Zn/Co-ZIF-8 materials as precursors for annealing. The synthesized materials have been characterized using XRD and XNES techniques to confirm the impregnation of cobalt ions into the ZnO structure. This research work is good and will be beneficial for the researchers working in this field. The present manuscript could be accepted after addressing the below comments/suggestions.

The authors thank Reviewer 2 for the high appreciation of the submitted work and helpful comments.

They are:

Please check the manuscript for typographical errors.

We double-checked all text to improve English.  

Some relevant references on metal organic frameworks are missing and should be cited in the main body of the manuscript

Meng, J., Liu, X., Niu, C., Pang, Q., Li, J., Liu, F., Liu, Z. and Mai, L., 2020. Advances in metal–organic framework coatings: versatile synthesis and broad applications. Chemical Society Reviews;

Verma, S., Baig, R.N., Nadagouda, M.N. and Varma, R.S., 2016. Titanium-based zeolitic imidazolate framework for chemical fixation of carbon dioxide. Green Chemistry, 18(18), pp.4855-4858;

Aguiar, L.W., Otto, G.P., Kupfer, V.L., Fávaro, S.L., Silva, C.T., Moisés, M.P., de Almeida, L., Guilherme, M.R., Radovanovic, E., Girotto, E.M. and Rinaldi, A.W., 2020. Simple, fast, and low-cost synthesis of MIL-100 and MIL-88B in a modified domestic microwave oven. Materials Letters, p.128127

Thank you, articles were quoted [8, 14]

Round 2

Reviewer 1 Report

Modifications have been performed by the authors and the manuscript can thus be accepted